# Adolescent-to-Parent Violence and Family Environment: The Perceptions of Same Reality?

**DOI:** 10.3390/ijerph16122215

**Published:** 2019-06-23

**Authors:** Izaskun Ibabe

**Affiliations:** College of Psychology, University of the Basque Country UPV/EHU, 20018 San Sebastián, Spain; izaskun.ibabe@ehu.es; Tel.: +34-943-01-5691

**Keywords:** child-to-parent violence, family environment, adolescents, parents, agreement between reports, corporal punishment, family discipline

## Abstract

The use of several sources of information (parents and children) is scarce in family studies. Child-to-parent violence (CPV) is still considered the most hidden and stigmatized form of family violence. One objective of this study was to analyze the prevalence of child-to-parent violence and perceptions of family environment as a function of the informant (parent or child), child’s sex, and parents’ sex in a community population. The study also aimed to analyze the predictive power of family conflict and aggressive family discipline in child-to-parent violence depending on the informant. A sample of 586 adolescents (49% boys, aged between 12 and 18) and their parents (40%) participated in the study. The Family Environment Scale and the Conflict Tactics Scales were administered. Results showed good consistency between adolescent reports and parent reports for physical CPV, but adolescents perceived worse family environments than their parents. Multiple regression models revealed that aggressive family discipline and family are important risk factors for CPV. Early intervention to prevent CPV is recommended, focused on promoting family relationships and avoiding harsh discipline practices. It is important that parents are able to ask for help when they need it.

## 1. Introduction

Family violence research over the last two decades has largely focused on child abuse and intimate partner violence. An issue that has received limited attention until recent years, however, is child-to-parent violence (CPV). In the current study, CPV is defined as violence, including physical and psychological violence, perpetrated by children or adolescents and directed toward their parents or caregivers. In this definition, the intention to control parents, which appears in the definitions of different experts (e.g., [1]), has been eliminated. Pereira et al. [2] have developed a practical definition which includes repeated violent behaviors of children towards their parents or caregivers when the child has a relationship of dependency with respect to the parent. Most studies have focused exclusively on community samples and children reports [3,4]. Findings based on adolescent reports indicate prevalence rates ranging from 7.2% to 22% for physical aggression, and from 65.8% to 93.5% for psychological aggression [5]. In community populations, in which levels of physical CPV are expected to be low, the results of some studies indicate that the differences between sons and daughters as perpetrators are non-existent or negligible, but girls are more likely to have higher levels of psychological CPV [3,6,7]. In legal contexts, where physical CPV can be more serious, sons are more frequent perpetrators than daughters [8]. In a recent systematic review, Simmons et al. [9] concluded that there was an overall trend towards gender symmetry of the perpetrator in self-reported physical CPV in community samples, but in offender samples, males accounted for 59–87%. However, community research studies report that daughters tend to use more psychological or verbal abuse toward parents than sons [3,4]. With respect to target gender, mothers are reported more frequently than fathers in offender samples [9]. For example, in a study by Ibabe and Jaureguizar [8], the most frequent victim of the aggression was the mother (97%) in both cases (when the perpetrator was the son or the daughter) and 83% of mother-abusers were sons. This could be due to the possible modeling of aggressive behavior when children witness intimate partner violence toward their mother, and it could also be because the children are frightened of their father. Although both parents are living in the household, the mother is still usually the victim [8]. However, some community studies found that fathers are as likely to be targeted as mothers [9]. This could mean that the victimization of the mother is related to the severity of the physical abuse involved.

The use of various sources of information (parents and children) in this type of study has been scarce. Recent research carried out in Spain by Calvete et al. [5] examined the consistency between parent reports and child reports when reporting on child-to-parent violence in a community sample. The results indicate that parents may underestimate the violence of which they are victims in a community population, although the effect size was small. The prevalence rates of psychological violence found in parent reports and adolescent reports were 93% and 89%, respectively, while the rates for physical violence were 22% and 11%. Some parents may blame themselves for their children treating them aggressively. In fact, society often interprets CPV as a failure in parental education and setting limits on their children [5], although other causes or explanations may be involved, such as children’s personal factors (problematic substance use or psychological disorders) [10]. However, in the most serious cases, the opposite may be obtained. On the one hand, in legal contexts, adolescents may report lower levels of violence than parents in order not to incriminate themselves, and as far as possible, to avoid any consequences their actions may have. On the other hand, in such situations, parents are driven to report child-to-parent violence because they do not feel able to control the situation and are seeking a solution. Both adolescents and parents are key sources of information about family environment, and adolescent and parent reports of family relationships rarely converge [11]. De los Reyes, Ohannessina, and Raez [12] indicate that studying discrepancies between adolescent and parent reports of their relationships provides relevant information about family functioning and adolescent development. Divergence between adolescent reports and parent reports on family relationships, with adolescent reports being the more negative, predicts more internalizing problems in adolescents [13]. It would be interesting to study associations between adolescent–parent discrepancies in family relationships and adolescent adjustment. 

One of the peculiarities of CPV is that parents seek protection from the same children they are responsible for protecting. Moreover, depending on the children’s age, occasional conflict is likely to occur between children and parents living together, and there may be situations where aggression and violence are accepted as normal [14]. From a psychological perspective, conflict in parent–adolescent relationships arises due to the need of adolescents to detach themselves emotionally from parents or parental figures [15]. Adolescents perceive that their privacy has been invaded [16]. Family conflict generally arises from disagreements over everyday, routine issues [17], and this type of conflict has the potential to lead to violence and abuse [18]. Among the many potential risk factors analyzed in different studies, parent-to-child violence and marital violence are highly relevant when considering community samples [6,10,19,20] and offender samples (child-to-parent offenders and other offenders) [21]. Numerous studies have shown that child abuse, exposure to inter-parental violence, and both in combination (e.g., dual exposure) increase a child’s risk of internalizing and externalizing outcomes in adolescence [22,23]. In fact, there is a great deal of empirical evidence for the hypothesis of bidirectionality of family violence [20,24,25]: children who have experienced parental abuse or have observed inter-parental violence tend to be more violent toward parents. In a community population, child aggression could represent a functional response to family strain or an attempt to cope with inadequate parental education [24]. Although there is a recognized relationship between family violence and child-to-parent violence, the mechanisms by which family violence affects child-to-parent violence are less well studied. In their review, Simmons et al. [9] concluded that exposure to family violence can have an indirect effect on child-to-parent violence by affecting social information processing and making an individual more vulnerable to violent behavior.

There is some evidence that family conflict is an important risk factor for psychological child-to-parent violence (e.g., [4]) and antisocial behavior [26,27]. The escalation of a parent–child conflict can lead to the use of aggressive discipline [28,29]. Three forms of aggressive discipline by parents can be distinguished: verbal aggression, physical aggression in the form of corporal punishment (CP), and physical abuse [29]. Aggressive discipline and child abuse can be considered variants of parental aggression, with most cases of child abuse emerging from the routine practice of physical discipline strategies [30]. In no case can aggressive parental discipline ever be justified as a way of controlling children. Aggressive parental disciplinary practices—including both physical punishment and harsh psychological discipline—can be viewed as clear forms of child maltreatment with significant consequences for both individuals and society [31]. Three forms of parental aggression toward children (psychological aggression, physical aggression, and physical abuse) represent different levels of severity on a continuum of parental aggression [32]. In this context, the use of physical force with the intention of causing a child pain or discomfort in order to correct or control the child’s behavior is usually termed corporal punishment (CP) [33], while parental aggression causing visible injuries to children constitutes “crossing the line” from discipline to abuse [34]. However, in the present study, corporal punishment is also considered to be a form of child abuse because, as Straus [35] has indicated, many child-abuse injuries are the result of corporal punishment. 

Inappropriate parental discipline strategies have negative consequences on the psychological adjustment of children [36,37]. Specifically, discipline strategies administered inconsistently [38], power-assertive disciplinary methods (where a child’s inappropriate behavior results in a negative consequence without explanation or justification) [4,6], and especially corporal punishment have been related to CPV [3,38,39]. In a longitudinal study, Hoyo–Bilbao et al. [3] found that corporal punishment is related to CPV regardless of the context in which it is used or the age and sex of the child. In this study, 43% of the adolescents admitted that their parents had used corporal punishment during the previous year. However, earlier studies with Spanish samples indicated higher prevalence rates of corporal punishment (approximately 60%) (for a review, see Reference [37]). Although the use of CP may be decreasing in recent years in Spain and other countries, educational interventions should be applied to reduce this type of family socialization practice. For those raised in violent homes, conflict can escalate from a disagreement to abuse or use of violence, rather than move toward a resolution of differences through talking and involvement. 

General strain theory [24,40] and coercion theory [41] emphasize the instrumental functions of child-to-parent violence. Coercion theory describes a process of mutual reinforcement during which parents inadvertently reinforce children’s difficult behaviors, which in turn elicits parent negativity, etc., until the interaction is discontinued when one of the members imposes him- or herself on the other. These theories assume that children respond violently to aversive family interactions, and such behavior serves to reduce strain presented by parents or family members. Forms of parental discipline aim to correct or monitor the child’s behavior, ensure short-term obedience, and promote the internalization of long-term values [39]. An emphasis on child well-being and child rights can also be seen in the prohibition of corporal punishment [42], in place in the majority of European countries. However, in the United States, the approval of hitting children and adolescents is ingrained in cultural norms and supported by legal statutes [43,44]. According to UNICEF (United Nations Children’s Fund) [45], around 6 in 10 children between the ages of 2 and 14 worldwide (almost a billion) are subjected to physical punishment by their caregivers on a regular basis. Diverse studies have found that more frequent use of corporal punishment is related to a higher prevalence of violence and endorsement of violence at a societal level [46]. Gámez–Guadix and Almendros [47] found that Spanish parents tended to apply more discipline strategies to their children than Anglo-Saxon parents.

### Objectives and Hypothesis

The present study aims to analyze whether the prevalence of child-to-parent violence and perception of family environment changes as a function of the informant (parent or child), child’s sex, and parents’ sex in a community population. We hypothesize that the prevalence rate of physical and psychological violence found in adolescent reports will be slightly higher than in parent reports, consistent with the findings of Calvete et al. [38]. In line with these findings, higher levels of family conflict and lower levels of family cohesion will be expected in adolescent reports compared to parent reports. Daughters are expected to claim a higher prevalence of psychological aggression than sons, based on study samples in non-clinical populations, but no difference is expected between sons and daughters in terms of physical aggression [3,4]. Parent sex is examined in aggressive family discipline because one parent could be more likely to use aggressive discipline practices than the other. The study also aims to evaluate the predictive power of family conflict and aggressive family discipline in child-to-parent violence depending on the informant (adolescent, father, and mother). In a structural equation model of child-to-parent violence based on family relationship, power-assertive discipline, and age of perpetrators, the explained variance was 57% [4]. In contrast to previous studies, which only contained adolescent reports, the present study will compare three models according to adolescent reports, mother reports, and father reports. Additionally, associations between adolescent–parent discrepancies in family conflict and cohesion will be explored with respect to child-to-parent violence (an indicator of adolescent maladjustment) and the children’s interest in their studies (an indicator of adolescent adjustment). The inclusion of adolescent and parent reports may contribute to a better understanding of child-to-parent violence, thereby improving adolescent health and well-being. 

## 2. Materials and Methods

### 2.1. Participants

A sample of 586 adolescents (49% boys, aged 12 to 18) and their parents (*n* = 398, aged 27 to 65) from eight schools in the Basque Country participated in the study. The sample of parents was composed of 161 pairs of parents, 60 single mothers and 16 single fathers. Forty-three percent of the students were from state (public) schools and the rest were from private schools. A total of 75% lived in nuclear families, 14% in single-mother families, with 7% in step-families, and 4% in extended or other types of family. 

### 2.2. Instruments and Variables

Socio-Demographic Data. A questionnaire was used to collect socio-demographic data on the children. Among the variables studied were sex, age, country of birth, family structure, educational level, and parental occupation. In order to measure the interest in studies, adolescents were required to indicate their level of interest in their studies on a Likert scale (1 = Very low; 5 = Very high).

Family Environmental Scale. (FES [48], Spanish version [49]). Two subscales of the Family Environmental Scale were administered to parents and children. These two subscales contained 18 items with a true/false response format: Cohesion (the degree of commitment and support family members offer each other, a sample item being: “In my family we really help and support each other”) and Conflict (the degree of explicitly expressed anger and conflict among family members). In general, the alpha reliability coefficients in this study were acceptable: adolescents (Cohesion α = 0.76; Conflict α = 0.60), mothers (Cohesion α = 0.65; Conflict α = 0.51), and fathers (Cohesion α = 0.65; Conflict α = 0.50).

Conflict Tactics Scale Child–Parents (CTS1 [50]). This scale contains 13 items from three dimensions: psychological violence (e.g., “Insult or threaten my father/mother” or “My child insulted me or threatened me”), mild physical violence, and serious physical violence. Parents and children were asked to take the previous year as a reference and use a scale with the following values: 0 (Never), 1 (Hardly ever), 2 (Sometimes), 3 (Frequently), and 4 (Almost always). In general, internal consistency results were quite acceptable in the sample of adolescents (psychological violence, α = 0.85; physical violence, α = 0.86), mothers (psychological violence, α = 0.75; physical violence, α = 0.67), and fathers (psychological violence, α = 0.71; physical violence, α = 0.49).

Dimensions of Discipline Inventory (DDI-C [42], Spanish adaptation [51]). Although this inventory measures four general dimensions, the present study only measured punitive discipline (corporal punishment and psychological aggression). Family discipline was assessed from the children’s point of view in their relationship with their father and mother. Items described different situations related to life and family upbringing, to which children were required to respond on a 5 point Likert scale, from 0 (Never) to 4 (Almost always). The subscales for corporal punishment (e.g., “How often did your father/mother shake or grab you to get your attention?”) and psychological aggression (e.g., “How often did your father/mother shout at you?”) had four questions each. In this study, the internal consistency for this dimension (α = 0.86) and two subscales (psychological aggression α = 0.81 and corporal punishment α = 0.82) was excellent. Furthermore, internal consistency alphas for corporal punishment by father (α = 0.70) and by mother (α = 0.74) were tolerable, as was the reliability of psychological aggression by father and (α = 0.77) by mother (α = 0.66) scales.

### 2.3. Procedure

This study was conducted in accordance with relevant international (American Psychological Association) and national (Código Deontológico del Psicólogo) ethics guidelines. The selection of the adolescent sample was performed using cluster sampling in secondary schools in the Basque Country. Eight schools participated after they had confirmed their availability and the willingness of their staff to take part in the research. Before collecting the data, head teachers were given detailed information about the objectives of the research in a one-hour presentation. A letter describing the study was sent to parents requesting that they indicate in writing whether or not they agreed to have their children participate in the research project. Participants were given guarantees of confidentiality and anonymity regarding their responses. In the classroom, the instructions for each questionnaire were read aloud before the students completed them. The questionnaires were administered during normal class time in one-hour sessions. Data collection was conducted during 2011, and administration time for the instruments was approximately 45 minutes. The order of presentation of the instruments was counterbalanced.

The procedure for collecting parent reports included the submission of an information sheet and evaluation protocol with an identification code linking them to their children. Once parents agreed to participate, they had to submit the completed questionnaire to the child’s tutor or by post within one week.

### 2.4. Data Analysis

Univariate data analysis was conducted using IBM SPSS Statistics version 23 (IBM Corporation, Bilbao, Spain). First, two variables on violence toward parents (physical and psychological violence) were dichotomized in terms of the response format (0 = Never, 1 = Hardly ever, 2 = Sometimes, 3 = Frequently, 4 = Almost always), with participants who chose option 1 or higher being considered as exercising or having exercised some form of violence towards their parents in the previous year. These dichotomous variables were used only to calculate the prevalence of two types of child-to-parent violence. Data analyses on the consistency of parent reports and child reports are presented in Table 1. These data analyses were done with 161 pairs of parents (161 fathers and 161 mothers) and their children. Family environment variables were analyzed by applying the paired sample *t*-test; in parent reports, the average of the father reports and the mother reports was calculated. The Wilcoxon test was applied to family environment, and differences were significant for family conflict (*z* = 2.75, *p* = 0.006) as well as for family cohesion (*z* = 3.05, *p* = 0.002). The results were not different from the *t*-test results. Differences between the prevalence of child-to-parent violence according to the sex of the children based on adolescent and parent reports were calculated. In order to examine whether victimization changes depending on parent sex, the perpetration rates of CPV toward fathers and toward mothers were calculated, taking into account adolescent reports and parent reports. The prevalence rates for corporal punishment by parents and psychological aggression by parents were computed, as well as the means comparisons depending on the parent’s sex and children’s sex. 

In order to analyze associations between adolescent–parent discrepancies in reports of family conflict and family cohesion, the patterns of informant discrepancies were analyzed. In terms of family conflict, when an adolescent report > parent report, it was considered a negative adolescent report, which was also the case for family cohesion when the adolescent report < parent report. Dichotomous variables (negative report (NR) versus non-negative report (NNR)) for family conflict and cohesion, respectively, were thus created. Means comparisons of child-to-parent violence and interest in studies according to adolescent reports were subsequently explored as a function of informant discrepancies.

Cramer’s V was calculated as a measurement of effect size for the chi-square test of independence. A small effect was reflected by values around 0.10, a medium effect by values around 0.30, and a large effect above 0.50. In order to study the differences in the means of family environment perception and frequency of child-to-parent violence, the Student’s *t*-test was applied with Cohen’s d for effect size.

Next, a correlation matrix was determined (see Table 2), in which ten quantitative variables were included. Finally, multiple regression analyses were carried out, entering family conflict and aggressive family discipline as independent variables, and female sex, children, and age as control variables with CPV (adolescent reports, mother reports, and father reports) as dependent variables (see Table 3). 

## 3. Results

### 3.1. Prevalence Rates of Child-to-Parent and Family Environment Perception

Table 1 shows prevalence rates of CPV and family environment averages by informant (adolescent or parent). The prevalence rate of child-to-parent violence found in adolescent reports was higher than in parent reports for psychological violence against the father (81% versus 76, χ^2^ (1, *N* = 168) = 4.85, *p* = 0.03, Cramer’s *V* = 0.17), while in the other types of child-to-parent violence the differences were not significant. Adolescents reported significantly higher scores in family conflict (*M* = 3.03) than their parents (*M* = 2.57), *t*(134) = 3.10, *p =* 0.002, *d* = 0.35, 95% CI (0.75, 0.17), and lower scores in family cohesion (*M* = 7.00) than their parents (*M* = 7.58), *t*(160) = 3.82, *p* < 0.001, *d* = 0.35, 95% *CI* (0.28, 0.88).

On the one hand, the differences between males and females as perpetrators of child-to-parent violence were analyzed. When adolescents were the informants, daughters were psychologically slightly more abusive toward their mothers (88%) than were sons (81%), χ2(1, *N* = 548) = 5.70, *p* = 0.017, Cramer’s *V* = 0.10. Taking into account means comparisons, daughters were psychologically slightly more abusive toward their mothers (*M* = 0.79) and fathers (*M* = 0.67) than were sons (*M* = 0.53 and *M* = 0.51), *t*(546) = 5.31, *p* < 0.001, *d* = 0.28, 95% *CI* (0.35, 0.16); *t*(508) = 3.14, *p* = 0.002, *d* = 0.46, 95% *CI* (0.26, 0.06). On the other hand, we also analyzed the differences between fathers and mothers as victims of child-to-parent violence. With respect to adolescent reports, the perpetration of psychological child-to-parent violence toward mothers (84%) was slightly more frequent than toward fathers (81%), χ2(1, *N* = 518) = 209.22, *p* < 0.001, Cramer’s *V* = 0.64. According to parent reports, the prevalence rate of psychological violence toward mothers (82%) was also higher than toward fathers (76%), χ2(1, *N* = 155) = 64.40, *p* < 0.001, Cramer’s *V* = 0.67. In means comparison, mother victimization (*M* = 0.64) was more frequent than father victimization (*M* = 0.59) for psychological violence, *t*(518) = 2.46, *p* = 0.014, *d* = 0.10, 95% *CI* (0.09, 0.01). When parents were the informants, the difference was almost significant, *p* = 0.059. Moreover, according to means comparisons of family environment measures, girls perceived a greater level of family conflict (*M* = 3.26) than boys (*M* = 2.69), *t*(568) = 3.76, *p* < 0.001, *d* = 0.31, 95% *CI* (0.86, 0.27), while fathers perceived (*M* = 2.63) a greater level of family conflict than mothers (*M* = 2.45), *t*(146) = 2.09, *p* = 0.038, *d* = 0.18, 95% *CI* (0.34, 0.01).

### 3.2. Relation between Child-to-Parent Violence and Other Family Variables

Aggressive family discipline was among the family variables studied. The prevalence rate for corporal punishment was 44% and 89% for psychological aggression by parents. Psychological aggressive discipline by the mother (*M* = 0.93) was more frequent than by father (*M* = 0.86), *t*(519) = 2.38, *p* = 0.018, *d* = 0.10, 95% *CI* (0.01, 0.14). Moreover, aggressive discipline by the mother was directed more frequently at daughters (*M* = 1.02) than sons (*M* = 0.86), *t*(549) = 2.25, *p* = 0.025, *d* = 0.19, 95% *CI* (0.30, 0.02).

To examine these relationships and influences, correlations between the family variables of the study were computed (see Table 2). The first three columns of correlations show that the CPV of different informants was related to aggressive family discipline (corporal punishment and psychological aggression) according to adolescent reports. It is evident that children’s perception of family conflict was related positively to aggressive family discipline. Moreover, the mother’s perception of family conflict was associated with aggressive family discipline by the mother.

### 3.3. Child-to-Parent Violence Models 

Table 3 shows the results of the multiple regression analysis testing the hypothesis that family conflict perception and aggressive family discipline are related to CPV. Three independent variables turned out to be significant predictors of CPV according to adolescent reports: corporal punishment by the father (*β* = 0.309, *p* < 0.001), family conflict (*β* = 0.245, *p* < 0.001), and psychological aggression by the mother (*β* = 0.214, *p* < 0.001). However, two variables were significant predictors of CPV according to mother reports: family conflict (*β* = 0.331, *p* < 0.001) and corporal punishment (*β* = 0.299, *p* < 0.001). Finally, Model 3 shows that family conflict (*β* = 0.303, *p* < 0.001) and psychological aggression (*β* = 0.218, *p* < 0.01) were significant predictors of CPV according to father reports.

Child-to-parent violence was higher in the negative report group (NR) of family conflict (*M* = 0.47) than the non-negative report group (NNR) (*M* = 0.22), *t*(73.02) = 3.70, *p* < 0.001, *d* = 0.68, 95% *CI* (0.38, 0.11), while interest in studies was lower in the NR group (*M* = 2.55) than the NNR group (*M* = 3.13), t(132) = 3.82, *p* < 0.001, *d* = 0.65, 95% *CI* (0.28, 0.89). Child-to-parent violence was higher in the NR group of family cohesion (*M* = 0.43) than the NNR group (*M* = 0.23), *t*(90.51) = 3.74, *p* < 0.01, *d* = 0.56, 95% *CI* (0.31, 0.08), when interest in studies was lower in the NR group (*M* = 2.64) than the NNR group(*M* = 2.99), *t*(158) = 2.38, *p* < 0.05, *d* = 0.37, 95% *CI* (0.06, 0.65).

## 4. Discussion

One objective of this study was to analyze whether the prevalence of child-to-parent violence and perception of family environment changed depending on the sex of the informant (parent or child) in a community population. As was expected, adolescent reports indicated higher prevalence rates of psychological CPV among daughters than sons, but no difference was found in physical aggression between sons and daughters. There are hardly any studies based on parent reports, but Calvete et al. [5] found that the prevalence rate in physical CPV of sons was higher than the prevalence rate of daughters, although the effect size was small (*V* Cramer = 0.16). These results are consistent with previous studies based on non-clinical populations and adolescent reports [3,4,6]. Gallagher [52] found that the survey data’s overall gender symmetry for CPV contrasted markedly with the almost three-to-one gender imbalance in studies with clinical, medical, police, and court data. Curiously, this pattern of results is similar to that identified in the intimate partner violence literature: gender symmetry in community samples and gender asymmetry in legal samples [53]. An important reason why the results from community-based surveys are different from those based on medical attendance, police or court files could be the severity of the physical abuse involved. 

Moreover, in this study, the target gender was examined for physical and psychological CPV as a function of the informant (adolescents and parents) with three categories (CPV toward mother only, CPV toward father only, and CPV toward both parents). A notable finding was that 77% of children direct psychological violence toward both parents. However, as was found in similar studies with a community population [19,25], there was no difference in the victimization of fathers or mothers. Although most studies on this family problem (based on the evidence from clinical and legal fields) unequivocally agree that mothers are more likely to be victims of abuse by their adolescent children [8,9], some studies focusing on serious CPV (e.g., parricide) have found that fathers are more likely to be victims of CPV than mothers (e.g., [54]). It would be interesting to discover whether the different findings regarding target gender are due to the severity of CPV or to different ways of measuring CPV. In any case, future research should consider gender-sensitive designs with reliable measures of two dimensions (i.e., frequency and severity) [9].

It was hypothesized that the prevalence rate of CPV in adolescent reports would be higher than in parent reports, but this hypothesis was only confirmed for psychological child-to-father violence. Moreover, as expected, family conflict scores in adolescent reports were higher than parent report scores, while the family cohesion scores of adolescent reports were lower than parent report scores. There were hardly any differences between adolescent reports and parent reports for CPV. However, the results of the present study indicated that parents underestimated the level of family conflict, perhaps due to the shame and blame their feelings of a negative family environment have on their own poor parenting practices. The perception of family environment varied by sex of the adolescent, with daughters indicating higher levels of family conflict and lower levels of family cohesion compared to sons. These findings are consistent with those of other studies [55,56]. According to Nelson et al. [56], perceived family characteristics differed by gender; female participants reported higher levels of family conflict and parental monitoring, as well as lower levels of family social support. Female adolescents may experience an increased tendency towards interpersonal connectedness and concern for the well-being of others [55], which may cause them to be more sensitive when observing family processes [52]. Additionally, it has been found that the perceived family environment in which adolescents are raised plays an important role in their adoption of health risk behaviors (e.g., increased risk for substance use disorders), this being particularly true for female adolescents with respect to family conflict and family social support [55,57]. Divergence between adolescent reports and parent reports on family relationships were analyzed. Negative reports of adolescents with respect to family conflict and family cohesion were associated with more child-to-parent violence, and less interest in studies. Measuring the distance between adolescent and parent reports of family relationships could help predict adolescent adjustment, but it is important to take into account the direction of adolescent–parent discrepancies. Particularly, when adolescents’ perceptions are more negative than parents’ perceptions, they appear to be relevant to adolescent functioning [13]. It is an emerging research field, and recent work supports the hypothesis that divergence between reports may not always predict negative adolescent outcomes [12].

In the present study, the corporal punishment prevalence rate was 44%, while for psychological aggression it was 89%. Aggressive discipline by parents may evoke feelings of fear, anxiety, and anger in children, and these emotions could interfere with a positive parent–child relationship, as was found in previous studies of corporal punishment [58]. In fact, aggressive family discipline was a valid predictor of children’s family conflict perceptions. The regression model of child-to-parent violence based on family conflict and aggressive family discipline based on adolescent reports explained 38% of the variance. It is an excellent and parsimonious statistical model. The results of the present study corroborated the relevance of aggressive family discipline and family conflict as risk factors for child-to-parent violence. Numerous studies had previously indicated that aggressive family discipline and family conflict are valid predictors of child-to-parent violence [3,4,35,36]. A longitudinal study with a national survey of male adolescents [31] analyzed whether child aggression represents a functional response to family strain, with results indicating a reciprocal relationship between parent-to-child violence and CPV, characterized by countervailing effects. Aggression by the adolescent may prove to be a partially successful means of combating family strain or negative intervention. Child-to-parent violence is, thus, not necessarily a form of pathology because it could be a survival response by children when their well-being is threatened [14]. 

The main limitation of this research is that it is a cross-sectional study. Cross-sectional data make it difficult to identify exactly how aggressive family discipline influences child-to-parent violence over time. It would be preferable to conduct longitudinal studies in clinical contexts. Aggressive family discipline was only measured through adolescent reports. The internal consistency of family environment subscales is lower than desirable (α ≥ 0.70) because this scale has positive and reversed items. When combinations of positive and reversed items are used in the same test, the reliability of the test is flawed [59]. However, it should be noted that father reports of physical CPV was low. A high proportion of missing data in father reports and mother reports was found, although missing data are frequent in studies based on multiple informant data. The fact that parents in the study represent a volunteer sample means that those parents who are most violent to their children were unlikely to participate. Such a skewed sample is, thus, likely to have had an impact on results. As sons and daughters were not from the same family, it was not possible to assume that we were analyzing the same family processes. Thus, we cannot be sure that daughters were more sensitive when observing family processes than sons.

Future research should obtain more detailed data on parental and child aggression in the events which occurred, including mild and severe forms of aggression, by using clinical or legal samples. Additionally, it would be interesting to study the directionality of interpersonal violence in child–parent relationships in order to know to what extent CPV is bidirectional or unidirectional violence.

## 5. Conclusions

In conclusion, there was consistency between adolescent reports and parent reports for CPV, and aggressive family discipline can be considered an important risk factor for child-to-parent violence. Nowadays, CPV is still considered the most hidden, misunderstood, and stigmatized form of family violence, and an early-help approach to stop problems from spiraling is recommended [60]. In answer to the question “why parents hide their child-to-parent violence situation instead of asking for help”, Concordia Gabinete [61] suggests five reasons for not requesting help: (1) parents do not really know the extent of the problem they have at home; (2) they are afraid; (3) they are ashamed to talk about the problem; (4) they have had bad experiences with professionals in the past; and (5) they do not want anything bad to happen to their son or daughter. The findings of this study could be applied to multiple disciplines and potentially lead to policy changes. Child and family services should take into account that parents may underestimate the levels of violence toward them and of family conflict, and in the future, it would be interesting to use a multi-informant approach to assess child-to-parent violence or family environment. Additionally, to prevent abusive family relations, including the occurrence of CPV, parents could benefit from training to reduce harsh discipline. However, when teenage and younger girls and boys use physical, psychological, emotional, and financial abuse and violence over time to the extent that parents live in fear of their child, parents need to be empowered to find a way out of such a situation.

## Figures and Tables

**Table 1 ijerph-16-02215-t001:** Prevalence rates of different types of child-to-parent violence (CPV) and family environment perception by informant.

Variables	Adolescent Reports %/*M*	Parent Reports %/*M*	χ2/*t*	Effect Size ^a^
*Psychological violence*				
Psychological violence child-to-mother	84.2%	81.9%	1.14	0.07
Psychological violence child-to-father	81.1%	75.7%	4.85 *	0.17
*Physical violence*				
Physical violence child-to-mother	7.0%	1.9%	2.05	0.10
Physical violence child-to-father	5.3%	2.3%	3.23	0.14
*Family environment*				
Family conflict (0–9)	3.03	2.57	3.03 *	0.34
Family cohesion (0–9)	7.00	7.58	3.83 **	0.35

Note: Zero tolerance criteria (when the response “Hardly ever” or more in terms of frequency was given in response to any item) was used to measure child-to-parent violence; * *p* < 0.05; ** *p* < 0.001. ^a^ Effect size was evaluated by Cramer’s *V* for chi-square analysis while Cohen’s d was calculated for *t*-test analysis.

**Table 2 ijerph-16-02215-t002:** Correlation matrix of family variables studied.

Variables	1	2	3	4	5	6	7	8	9
**Child-to-parent violence**									
1. CPV Adolescent reports	*-*								
2. CPV Mother reports	0.406 **	*-*							
3. CPV Father reports	0.302 **	0.616 **	*-*						
**Family conflict**									
4. Family conflict Adolescent reports	0.447 **	0.368 **	0.196 **	*-*					
5. Family conflict Mother reports	0.091	0.405 **	0.291 **	0.254 **	*-*				
6, Family conflict Father reports	0.060	0.279 **	0.327 **	0.054	0.551 **	-			
**Aggressive family discipline** Adolescent reports									
7. Corporal punishment by mother	0.428 **	0.324 **	0.196 **	0.444 **	0.200 **	0.136	-		
8. Corporal punishment by father	0.455 **	0.255 **	0.227 **	0.367 **	0.013	0.126	0.590 **	-	
9. Psychological aggression by mother	0.445 **	0.301 **	0.250 **	0.442 **	0.274 **	0.090	0.580 **	0.295 **	-
10. Psychological aggression by father	0.421 **	0.243 **	0.315 **	0.340 **	0.038	0.107	0.304 **	0.572 **	0.611 **

* *p* < 0.05; ** *p* < 0.0l; *** *p* < 0.001.

**Table 3 ijerph-16-02215-t003:** Multiple regression models for child-to-parent violence depending on informant.

Variables	Model 1 CPV Adolescent Reports	Model 2 CPV Mother Reports	Model 3 CPV Father Reports
**Family conflict**			
Family conflict Adolescent reports	0.245 ***	-	-
Family conflict Mother reports	-	0.331 **	-
Family conflict Father reports	-	-	0.303 ***
**Aggressive family discipline** Adolescent reports			
Corporal punishment by mother	-	0.299 ***	-
Corporal punishment by father	0.309 ***	-	-
Psychological aggression by mother	0.214 ***	-	0.218 **
Psychological aggression by father	-	-	-
**Socio-demographic variables**			
Female sex children	0.137 **		
Age of children	0.122 **		
Model *F*	50.43 ***	25.57 ***	12.25 ***
*R* ^2^	0.377	0.241	0.139

* *p* < 0.05; ** *p* < 0.0l; *** *p* < 0.001.

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
