# Peer review of "Adolescent-to-Parent Violence and Family Environment: The Perceptions of Same Reality?"

_ijerph, 2019, doi:10.3390/ijerph16122215_

Round 1

Reviewer 1 Report

Abstract

The author writes about a predictive model for child to parent violence, yet the research is based on cross-sectional data. It can not be predictive as you do not know the direction of the relationships.

Within the abstract mention is made that "daughters were more sensitive when observing family processes than sons". Were these children from the same family? If not, how can you assume you are reviewing the same family processes.

Introduction

I find it interesting that the author briefly mentions (1) that the mother is the most frequent target of CPV and (2) "marital violence" is a risk factor for CPV, yet there is no expansion of this idea to the possibility that because violence against the mother has almost been normalised within the family environment (where intimate partner violence exists) or because the children are frightened of the father, the mother becomes the target of the child's violence. 

What is the "role of the fathers and mothers in the socialisation of their children"? Is this the authors indicating that one parent is more likely to use aggressive discipline practices than the other? If so, it would be useful for this to be stated clearly.

Discussion

Why such a big focus on gender symmetry? The aim of the investigation was to determine concordance between child and adult reporting of CPV (according to the abstract, although I note there is more of a focus on gender symmetry in the methods). There are a number of reasons why the results from community-based surveys are different from those based on medical attendance, police or court files (for example, the severity of the physical abuse, physical dynamics involved such as the size / weight of the child compared with the parent). The prevalence estimates showed that, where only one parent was targeted, mothers were three times more likely to be targeted than fathers. That this was not a significant difference could be the result of the investigation being under-powered to detect such differences. Further, there has been no exploration if boys or girls reported the use of violence more often, if their parents corroborated their reporting, the duration that violence has been used by either the boys or the girls, or the level of fear experienced by the parents. The lack of context makes it difficult to report "gender symmetry"  

Conclusion

You didn't measure directionality. So how can you make a concluding statement on directionality?

Author Response

Abstract. The author writes about a predictive model for child to parent violence, yet the research is based on cross-sectional data. It can not be predictive as you do not know the direction of the relationships.

“Predictive model” term has been removed from all sentences of the manuscript.

Page 1: Multiple regression models revealed that aggressive family discipline and family are important risk factors for CPV.

Page 4: In a SEM model of child-to-parent violence based on family relationship, power-assertive discipline and age of perpetrators, the explained variance was 57% [4].

Page 13: The regression model of child-to-parent violence based on family conflict and aggressive family discipline based on adolescent reports explained 38% of variance. It is an excellent and parsimonious statistical model.

Abstract. Within the abstract mention is made that "daughters were more sensitive when observing family processes than sons". Were these children from the same family? If not, how can you assume you are reviewing the same family processes.

This sentence has been deleted from abstract. A new sentence has been included as limitation (Page 13):

As sons and daughters were not from the same family, it is not possible to assume that we are analyzing the same family processes. Thus, we cannot be sure that daughters were more sensitive when observing family processes than sons.

Introduction. I find it interesting that the author briefly mentions (1) that the mother is the most frequent target of CPV and (2) "marital violence" is a risk factor for CPV, yet there is no expansion of this idea to the possibility that because violence against the mother has almost been normalised within the family environment (where intimate partner violence exists) or because the children are frightened of the father, the mother becomes the target of the child's violence. 

The mother as more frequent target of CPV has been extended (Page 2):

With respect to target gender, mothers are reported more frequently than fathers in offender samples [9]. For example, in a study by Ibabe and Jaureguizar [8], the most frequent victim of the aggression was the mother (97%) in both cases (when the perpetrator was the son or the daughter) and 83% of mother-abusers were sons. This could be due to possible modeling of aggressive behavior when children witness intimate partner violence toward their mother, and it could also be because the children are frightened of their father. Although both parents are living in the household, the mother is still usually the victim [8]. However, some community studies found that fathers are as likely to be targeted as mothers [9]. This could mean that the victimization of the mother is related to the severity of the physical abuse involved.

Marital violence as a risk factor has been extended (Pages 2-3):

Numerous studies have shown that child abuse, exposure to inter-parental violence, and both in combination (e.g., dual exposure) increase a child’s risk of internalizing and externalizing outcomes in adolescence [22, 23]. In fact, there is a great deal of empirical evidence for the hypothesis of bidirectionality of family violence [20, 24, 25, 26]: Children who have experienced parental abuse or have observed inter-parental violence tend to be more violent toward parents. In a community population, child aggression could represent a functional response to family strain or an attempt to cope with inadequate parental education [24]. Although there is a recognized relationship between family violence and child-to-parent violence, the mechanisms by which family violence affects child-to-parent violence are less well studied. In their review Simmons et al. [9] concluded that exposure to family violence can have an indirect effect on child-to-parent violence by affecting social information processing and making an individual more vulnerable to violent behavior.

Introduction. What is the "role of the fathers and mothers in the socialisation of their children"? Is this the authors indicating that one parent is more likely to use aggressive discipline practices than the other? If so, it would be useful for this to be stated clearly.

The sentence has been changed (Page 4):

Parent sex is examined in aggressive family discipline because one parent could be more likely to use aggressive discipline practices than the other.

Additional analysis has been conducted to compare aggressive discipline practices as a function of parent’s sex and of children’s sex (Page 9):

Aggressive family discipline was among the family variables studied. The prevalence rate for corporal punishment was 44% and 89% for psychological aggression by parents. Psychological aggressive discipline by mother (M = .93) was more frequent than by father (M = .86), t(519) = 2.38, p = .018, d = .10, 95 % CI [.01, .14]. Moreover, aggressive discipline by mother was directed more frequently at daughters (M = 1.02) than sons (M = .86), t(549) = 2.25, p = .025, d = .19, 95 % CI [.30, .02].

Results. Further, there has been no exploration if boys or girls reported the use of violence more often, if their parents corroborated their reporting, the duration that violence has been used by either the boys or the girls, or the level of fear experienced by the parents. The lack of context makes it difficult to report "gender symmetry"

The text about prevalence rates of CPV as a function of parent’s sex and children’s sex has been reorganized and simplified. Parent reports of child-to-parent violence indicate the frequency of violent behavior toward parents by children’s sex (Page 7):

On the one hand, the differences between males and females as perpetrators of child-to-parent violence were analyzed. When adolescents were the informants, daughters were psychologically slightly more abusive toward their mothers (88%) than were sons (81%), χ2(1, N = 548) = 5.70, p = .017, Cramer’s V = .10. Taking into account means comparisons, daughters were psychologically slightly more abusive toward their mothers (M = .79) and fathers (M = .67) than were sons (M = .53 and M = .51), t(546) = 5.31, p < .001, d = .28, 95 % CI [.35, .16];  t(508) = 3.14, p = .002, d = .46, 95 % CI [.26, .06]. On the other hand, we also analyzed the differences between fathers and mothers as victims of child-to-parent violence. With respect to adolescent reports, the perpetration of psychological child-to-parent violence toward mothers (84%) was slightly more frequent than toward fathers (81%), χ2(1, N = 518) = 209.22, p <.001, Cramer’s V = .64. According to parent reports, the prevalence rate of psychological violence toward mothers (82%) was also higher than toward fathers (76%), χ2(1, N = 155) = 64.40, p < .001, Cramer’s V = .67. In means comparison, mother victimization (M = .64) is more frequent than father victimization (M = .59) for psychological violence, t(518) = 2.46, p = .014, d = .10, % CI [.09, .01]. When parents were the informants, the difference was almost significant, p = .059. Moreover, according to means comparisons of family environment measures, girls perceived a greater level of family conflict (M = 3.26) than boys (M = 2.69), t(568) = 3.76, p < .001, d = .31, 95 % CI [.86, .27], while fathers perceived (M = 2.63) a greater level of family conflict than mothers (M = 2.45), t(146) = 2.09, p = .038, d = .18, 95 % CI [.34, .01].

Discussion. Why such a big focus on gender symmetry? The aim of the investigation was to determine concordance between child and adult reporting of CPV (according to the abstract, although I note there is more of a focus on gender symmetry in the methods).

The first objective in the abstract has been improved to include gender symmetry comparisons (Page 1):

One objective of this study was to analyze the prevalence of child-to-parent violence and perception of family environment as a function of informant (parent or child), child sex and parent sex in a community population.

The text about prevalence rates of CPV as a function of parent’s sex and children’s sex has been reorganized and simplified in Results (Page 7):

On the one hand, the differences between males and females as perpetrators of child-to-parent violence were analyzed. When adolescents were the informants, daughters were psychologically slightly more abusive toward their mothers (88%) than were sons (81%), χ2(1, N = 548) = 5.70, p = .017, Cramer’s V = .10. Taking into account means comparisons, daughters were psychologically slightly more abusive toward their mothers (M = .79) and fathers (M = .67) than were sons (M = .53 and M = .51), t(546) = 5.31, p < .001, d = .28, 95 % CI [.35, .16];  t(508) = 3.14, p = .002, d = .46, 95 % CI [.26, .06]. On the other hand, we also analyzed the differences between fathers and mothers as victims of child-to-parent violence. With respect to adolescent reports, the perpetration of psychological child-to-parent violence toward mothers (84%) was slightly more frequent than toward fathers (81%), χ2(1, N = 518) = 209.22, p <.001, Cramer’s V = .64. According to parent reports, the prevalence rate of psychological violence toward mothers (82%) also was higher than toward fathers (76%), χ2(1, N = 155) = 64.40, p < .001, Cramer’s V = .67. In means comparison, mother victimization (M = .64) is more frequent than father victimization (M = .59) for psychological violence, t(518) = 2.46, p = .014, d = .10, % CI [.09, .01]. When parents were the informants, the difference was almost significant, p = .059. Moreover, according to means comparisons of family environment measures, girls perceived a greater level of family conflict (M = 3.26) than boys (M = 2.69), t(568) = 3.76, p < .001, d = .31, 95 % CI [.86, .27], while fathers perceived (M = 2.63) a greater level of family conflict than mothers (M = 2.45), t(146) = 2.09, p = .038, d = .18, 95 % CI [.34, .01].

There are a number of reasons why the results from community-based surveys are different from those based on medical attendance, police or court files (for example, the severity of the physical abuse, physical dynamics involved such as the size / weight of the child compared with the parent).

The text has been improved (Page 10):

These results are consistent with previous studies based on non-clinical populations and adolescent reports [3, 4, 6]. Gallagher [52] found that the survey data’s overall gender symmetry for CPV contrasted markedly with the almost three-to-one gender imbalance in studies with clinical, medical, police and court data. Curiously, this pattern of results is similar to that identified in the intimate partner violence literature: gender symmetry in community samples and gender asymmetry in legal samples [54]. An important reason why the results from community-based surveys are different from those based on medical attendance, police or court files could be the severity of the physical abuse involved.

The prevalence estimates showed that, where only one parent was targeted, mothers were three times more likely to be targeted than fathers. That this was not a significant difference could be the result of the investigation being under-powered to detect such differences.

The text has been revised (Page 7):

On the other hand, we also analyzed the differences between fathers and mothers as victims of child-to-parent violence. With respect to adolescent reports, the perpetration of psychological child-to-parent violence toward mothers (84%) was slightly more frequent than toward fathers (81%), χ2(1, N = 518) = 209.22, p <.001, Cramer’s V = .64. According to parent reports, the prevalence rate of psychological violence toward mothers (82%) also was higher than toward fathers (76%), χ2(1, N = 155) = 64.40, p < .001, Cramer’s V = .67. In means comparison, mother victimization (M = .64) is more frequent than father victimization (M = .59) for psychological violence, t(518) = 2.46, p = .014, d = .10, % CI [.09, .01]. When parents were the informants, the difference was almost significant, p = .059.

Conclusion. You didn't measure directionality. So how can you make a concluding statement on directionality?

The following sentence has been removed (Page 12):

There may be different profiles of CPV: abusive parents, victimized parents and bidirectional violence.

Reviewer 2 Report

The authors have answered some of the points raised in an earlier review, but not all of them. I like that SEM is not currently used. However, I still have significant issues with the authors using the word directionality, as it is misleading. Also, it is still difficult to read and interpret the tables. As before, I suggest examining discrepancy as a variable on its own. 

In addition, the authors seem to have added pie charts, which is difficult to understand, as they are generally considered difficult to interpret. I suggest removing them or making different graphs. 

Author Response

Reviewer 2

I still have significant issues with the authors using the word directionality, as it is misleading.

The text has been removed (Page 13):

In previous studies, the bidirectionality of family violence in child-to-parent violence has been mentioned [16], and the methodology used to assess this has used multiple regression analysis or structural equation modeling. These are ways to analyze the predictive capacity of child victimization (parent-to-child violence and marital violence exposure) on child-to-parent violence. The expression of directionality of violence perpetrated toward a partner has been investigated, with a distinction drawn between unidirectional and bidirectional violence [46, 52]. This categorization may be applied to child-to-parent violence: unidirectional parent-to-child violence, unidirectional child-to-parent violence and bidirectional. Each family member could be a victim, a perpetrator, or both victim and perpetrator. Bidirectional violence does not presuppose that parent and child are equally violent. It may not be symmetrical when reasons and consequences are taken into account [16]. When physical injury is considered, one member of the family could perpetrate more severe physical violence than another. Sometimes, family violence may be defensive. In general, CPV may have similarities with other interpersonal violence (intimate partner violence, adolescent antisocial behavior or peer abuse) and also potentially unique characteristics [9]. In future studies, it would be very interesting to analyze the prevalence rate of directionality of violence in parent-child relationships.

It is still difficult to read and interpret the tables.

The text about prevalence rates of CPV as a function of parent’s sex and children’s sex has been reorganized and simplified (Page 7):

On the one hand, the differences between males and females as perpetrators of child-to-parent violence were analyzed. When adolescents were the informants, daughters were psychologically slightly more abusive toward their mothers (88%) than were sons (81%), χ2(1, N = 548) = 5.70, p = .017, Cramer’s V = .10. Taking into account means comparisons, daughters were psychologically slightly more abusive toward their mothers (M = .79) and fathers (M = .67) than were sons (M = .53 and M = .51), t(546) = 5.31, p < .001, d = .28, 95 % CI [.35, .16];  t(508) = 3.14, p = .002, d = .46, 95 % CI [.26, .06]. On the other hand, we also analyzed the differences between fathers and mothers as victims of child-to-parent violence. With respect to adolescent reports, the perpetration of psychological child-to-parent violence toward mothers (84%) was slightly more frequent than toward fathers (81%), χ2(1, N = 518) = 209.22, p <.001, Cramer’s V = .64. According to parent reports, the prevalence rate of psychological violence toward mothers (82%) also was higher than toward fathers (76%), χ2(1, N = 155) = 64.40, p < .001, Cramer’s V = .67. In means comparison, mother victimization (M = .64) is more frequent than father victimization (M = .59) for psychological violence, t(518) = 2.46, p = .014, d = .10, % CI [.09, .01]. When parents were the informants, the difference was almost significant, p = .059. Moreover, according to means comparisons of family environment measures, girls perceived a greater level of family conflict (M = 3.26) than boys (M = 2.69), t(568) = 3.76, p < .001, d = .31, 95 % CI [.86, .27],  while fathers perceived (M = 2.63) a greater level of family conflict than mothers (M = 2.45), t(146) = 2.09, p = .038, d = .18, 95 % CI [.34, .01].

As before, I suggest examining discrepancy as a variable on its own.

New information about discrepancies between adolescent and parent reports regarding family relationships has been added. The different patterns of discrepancy on family relationship could predict adolescent adjustment. This is a new field research:

Introduction (Page 2)

 Both adolescents and parents are key sources of information about family environment, and adolescent and parent reports of family relationships rarely converge [11]. De los Reyes, Ohannessina and Raez [12] indicate that studying discrepancies between adolescent and parent reports of their relationships provides relevant information about family functioning and adolescent development. Divergence between adolescent reports and parent reports on family relationships, with adolescent reports being the more negative, predicts more internalizing problems in adolescents [13]. It would be interesting to study associations between adolescent-parent discrepancies in family relationship and adolescent adjustment.

Objectives (Page 4)

Additionally, associations between adolescent-parent discrepancies in family conflict and cohesion will be explored with respect to child-to-parent violence (an indicator of adolescent maladjustment) and the interest in studies variable (an indicator of adolescent adjustment). 

Data analyses (Page 6)

In order to analyze associations between adolescent-parent discrepancies in reports of family conflict and family cohesion, the patterns of informant discrepancies are analyzed. In terms of family conflict, when adolescent report > parent report, it is considered negative adolescent report, which is also the case for family cohesion when adolescent report < parent report. Dichotomous variables (negative report –NR- vs. non-negative report –NNR) for family conflict and cohesion respectively are thus created. Means comparisons of child-to-parent violence and interest in studies according to adolescent reports are subsequently explored as a function of informant discrepancies.

Results (Page 11)

            Child-to-parent violence was higher in the negative-report group (NR) of family conflict (M = .47) than the non-negative report group (NNR) (M = .22), t(73.02) = 3.70, p < .001, d = .68, 95 % CI [.38, .11], while interest in studies was lower in the NR group (M = 2.55) than the NNR group(M = 3.13), t(132) = 3.82, p < .001, d = .65, 95 % CI [.28, .89]. Child-to-parent violence was higher in the NR group of family cohesion (M = .43) than the NNR group (M = .23), t(90.51) = 3.74, p < .01, d = .56, 95 % CI [.31, .08], when interest in studies was lower in the NR group (M = 2.64) than the NNR group(M = 2.99), t(158) = 2.38, p < .05, d = .37, 95 % CI [.06, .65].            

Discussion (Page 12)

Divergence between adolescent reports and parent reports on family relationships were analyzed. Negative report of adolescents with respect to family conflict and family cohesion were associated with more child-to-parent violence, and less interest in studies. Measuring the distance between adolescent and parent reports of family relationships could help predict adolescent adjustment, but it is important taking into account the direction of adolescent–parent discrepancies. Particularly when adolescents’ perceptions are more negative than parents’ perceptions, appears to be relevant to adolescent functioning [13]. It is an emerging research field, and recent work supports the hypothesis that divergence between reports may not always predict negative adolescent outcomes [12].  

The authors seem to have added pie charts, which is difficult to understand, as they are generally considered difficult to interpret. I suggest removing them or making different graphs.

The charts have been removed, and the text has been improved (Page 7):

On the other hand, we also analyzed the differences between fathers and mothers as victims of child-to-parent violence. With respect to adolescent reports, the perpetration of psychological child-to-parent violence toward mothers (84%) was slightly more frequent than toward fathers (81%), χ2(1, N = 518) = 209.22, p <.001, Cramer’s V = .64. According to parent reports, the prevalence rate of psychological violence toward mothers (82%) also was higher than toward fathers (76%), χ2(1, N = 155) = 64.40, p < .001, Cramer’s V = .67. In means comparison, mother victimization (M = .64) is more frequent than father victimization (M = .59) for psychological violence, t(518) = 2.46, p = .014, d = .10, % CI [.09, .01]. When parents were the informants, the difference was almost significant, p = .059.

Other changes

The variable interest in studies has been included in relationship to analyze negative report (Page 4):

Socio-Demographic Data. A questionnaire was used to collect socio-demographic data on the children. Among the variables studied were sex, age, country of birth, family structure, educational level, and parental occupation. In order to measure the interest in studies, adolescents were required to indicate their level of interest in their studies on a Likert scale (1 = Very low; 5 = Very high).

Round 2

Reviewer 2 Report

This revision is much improved, the authors have clearly revised the paper well.